# Comparative Transcriptome Analysis of Developing Seeds and Silique Wall Reveals Dynamic Transcription Networks for Effective Oil Production in *Brassica napus* L.

**DOI:** 10.3390/ijms20081982

**Published:** 2019-04-23

**Authors:** Muhammad Shahid, Guangqin Cai, Feng Zu, Qing Zhao, Muhammad Uzair Qasim, Yueyun Hong, Chuchuan Fan, Yongming Zhou

**Affiliations:** 1National Key Laboratory of Crop Genetic Improvement, Huazhong Agricultural University, Wuhan 430070, China; m.shahid@webmail.hzau.edu.cn (M.S.); gqcai@mail.hzau.edu.cn (G.C.); zufeng1984@163.com (F.Z.); zhaoqing123@webmail.hzau.edu.cn (Q.Z.); uzairqasim1149@yahoo.com (M.U.Q.); hongyy@mail.hzau.edu.cn (Y.H.); fanchuchuan@mail.hzau.edu.cn (C.F.); 2Industrial Crops Institute, Yunnan Academy of Agricultural Sciences, Kunming 650225, China

**Keywords:** *Brassica napus*, RNA sequencing, transcriptomic profiling, oil content, lipid, metabolism

## Abstract

Vegetable oil is an essential constituent of the human diet and renewable raw material for industrial applications. Enhancing oil production by increasing seed oil content in oil crops is the most viable, environmentally friendly, and sustainable approach to meet the continuous demand for the supply of vegetable oil globally. An in-depth understanding of the gene networks involved in oil biosynthesis during seed development is a prerequisite for breeding high-oil-content varieties. Rapeseed (*Brassica napus*) is one of the most important oil crops cultivated on multiple continents, contributing more than 15% of the world’s edible oil supply. To understand the phasic nature of oil biosynthesis and the dynamic regulation of key pathways for effective oil accumulation in *B. napus*, comparative transcriptomic profiling was performed with developing seeds and silique wall (SW) tissues of two contrasting inbred lines with ~13% difference in seed oil content. Differentially expressed genes (DEGs) between high- and low-oil content lines were identified across six key developmental stages, and gene enrichment analysis revealed that genes related to photosynthesis, metabolism, carbohydrates, lipids, phytohormones, transporters, and triacylglycerol and fatty acid synthesis tended to be upregulated in the high-oil-content line. Differentially regulated DEG patterns were revealed for the control of metabolite and photosynthate production in SW and oil biosynthesis and accumulation in seeds. Quantitative assays of carbohydrates and hormones during seed development together with gene expression profiling of relevant pathways revealed their fundamental effects on effective oil accumulation. Our results thus provide insights into the molecular basis of high seed oil content (SOC) and a new direction for developing high-SOC rapeseed and other oil crops.

## 1. Introduction

Vegetable oil is an essential component of food and crucial for human health, and a renewable resource for biodiesel and other industrial raw materials [1,2]. With seed production of 67.45 million metric tons and oil production of 26.38 million metric tons (2016), oilseed rape (OSR; *Brassica napus* (2*n* = 38)) is the second largest oilseed crop and third largest vegetable oil contributor, accounting for more than 15% of the world’s vegetable oil production [3]. The demand for vegetable oil has risen sharply and is expected to reach 200 billion kg by 2030 [2,4]. As crop growing areas are declining due to climate change and urbanization, increasing the oil yield by enhancing the oil content is an environmentally friendly, economical, viable, and sustainable approach. A 1% increase in the seed oil content of rapeseed is equivalent to a 2.3–2.5% increase in seed yield [5]. A steady enhancement of oil content through regulation of seed oil biosynthesis and accumulation in oilseed rape has important economic significance [2,6].

The major seed reserves in *B. napus* are carbohydrate, protein, and triacylglycerol (TAG), among which oil accumulates in the form of TAG [7,8,9]. Oil biosynthesis is initiated once embryo morphogenesis is achieved [10]. In *B. napus*, oil accumulation usually starts 17 to 24 days after pollination (DAP) [11], and 20–45 DAP has been observed a key stage for oil biosynthesis and accumulation [12].

In OSR, silique is a composite organ that comprises seeds and silique wall (SW) [13]. SW protects developing seeds inside, and provides the source to sink transport links to accumulate storage reserves in seeds and provide nutrients. Moreover, SW also acts as the major photosynthetic organ after flowering, when leaf senescence and abscission occur. Chlorenchyma cells in the mesocarp of SW are rich in chloroplast, which endows the SW with photosynthesis capability [14]. The rate of carbon dioxide fixation in SW can be high as 35% of that in leaves, which is significantly higher than the fixation by pod walls of legumes and chickpea (<5%) [15,16,17].

On the other hand, the metabolic process in seeds is very different from that in other plant tissues. During fatty acid (FA) synthesis in the seed, rubisco (ribulose-1,5-bisphosphate carboxylase-oxygenase) enhances the carbon use efficacy by skipping the Calvin cycle [18]. Since the seed storage reserves are mainly determined by intrinsic metabolic pathways and the availability of assimilates and nutrients directly or indirectly provided by the SW, a better understanding of the coordinated mechanisms between these two important organs is of critical importance for the enhancement of seed oil biosynthesis and accumulation.

Arabidopsis thaliana possesses more than 700 acyl lipid metabolism genes [19], while *B. napus* has 1898 lipid-related genes, 1097 and 1132 from A and C genomes, respectively. Oilseed rape *B. napus* has a fantastic reservoir of oil biosynthesis genes, 2.2% of total genes highly conserved for oil biosynthesis [20]. There are twice as many oil biosynthesis genes in *B. napus* than in soybean and oil palm [21,22]. Understanding the regulation of lipid metabolism at the molecular level is vital for genetic engineering to increase the oil yield or modify oil composition.

Expression dynamics of oil biosynthesis genes present in OSR have an unusual pattern that is different from all other plants. Many allied pathways play a role in the lipid biosynthesis pathway. So, in order to understand oil biosynthesis and accumulation, in-depth knowledge of the gene networks involved is critical [23]. Contrasting observations of temporal changes in the expression pattern of oil biosynthesis genes during seed development have been reported in previous studies on *B. napus*. A bell-shaped temporal expression pattern was observed in References [24,25], while a continuous decline in relative abundance of expressed sequence tags for many oil biosynthesis enzymes during seed development in *B. napus* was reported in Reference [4].

*Brassica napus* (genome AACC, 2*n* = 38) is allopolyploid and descends from diploids Asian cabbage (*B. rapa*, genome AA, 2*n* = 20) and Mediterranean cabbage (*B. oleracea*, genome CC, 2*n* = 18). It is a close relative of A. thaliana and considered as an ideal plant for manipulating information from the model plant [26]. However, it is difficult to establish a simple one-to-one relationship with Arabidopsis due to frequent duplication and rearrangements in the *B. napus* genome [27]. Many studies have reported attempts to identify the biosynthesis process gene in various crops by transcriptome analysis.

Although earlier studies showed that lipid biosynthesis pathways in Brassica seeds are similar to Arabidopsis, there are very few references available on transcriptomic regulation of oil accumulation in seeds and silique wall across the major seed development stages in the crop, especially in field conditions. In this study, we performed multistage comparative transcriptomic profiling of two inbred lines that differ significantly in oil content to understand the dynamic regulation of effective oil biosynthesis and accumulation. Our results provide new insights for a better understanding of how to achieve more effective oil production by manipulation of core pathways and genes involved.

## 2. Results

### 2.1. Dynamics of Oil Accumulation and Carbohydrates in HOC and LOC Lines

Two *B. napus* inbred lines with significant variation of oil content (~13%) were characterized in two consecutive years and two locations (Wuhan and Chongqing). The high-oil-content line (HOCL) accumulated more oil compared to the low-oil-content line (LOCL) across years and locations, suggesting that the difference in oil content between the two genotypes is highly stable (Figure 1A1).

To obtain a comprehensive comparison of oil accumulation dynamics in seed and SW tissues between high- and low-oil lines during seed development, we quantified the oil content in both tissues with samples collected at 11, 16, 23, 30, 37, and 44 DAP. After very slow changes in the first two stages (11 and 16 DAP), the seed oil content (SOC) of both lines increased sharply from 23 to 44 DAP, followed by a slight decrease after 44 DAP (Figure 1A2). It is interesting to note that the difference in SOC between the HOCL and LOCL existed at the very beginning, while the difference became more striking after 23 DAP in the HOCL. It seemed that the major portion of the oil was synthesized and accumulated in the developmental period from 23 to 44 DAP and the highly significant difference between the HOCL and LOCL arose in this stage of development (Figure 1A2).

In contrast with the dynamic pattern of oil content in seeds, the oil content in SW exhibited a continuous decline from the start point of 11 DAP to 37 DAP. The difference in the two genotypes was marginal and the curves intersected between the points 16 and 23 DAP. Overall, there were no significant differences in SW between the HOCL and LOCL except at 23 DAP (Figure 1A3).

To acquire an overview of carbohydrate-related phenotypic changes during seed and SW development, the same seed and SW samples for RNA-Seq and oil content quantification were used for carbohydrate analysis. Glucose, sucrose, and starch contents were measured at all six developmental stages (11, 16, 23, 30, 37, and 44 DAP) in seed and SW of both the HOCL and LOCL (Figure 1B). The HOCL had a significantly higher amount of glucose in seed and SW at all stages except 37 and 44 DAP in seed tissue, where the difference was nonsignificant. Starch content in the HOCL was also significantly higher at all stages except 11, 30, and 44 DAP in seed and 11 DAP in SW, which showed nonsignificant variance. Similarly, starch content was also significantly higher in the HOCL at all developmental stages of seed and SW except 44 DAP (Figure 1B1–B6). Overall, significantly higher contents of glucose, sucrose, and starch were observed in the HOCL as compared to LOCL during the key phase of oil biosynthesis from 23 to 44 DAP in seed and SW tissues. The results support the hypothesis that carbon flux during the key oil biosynthesis phase is the foundation for effective oil accumulation.

### 2.2. RNA Sequencing and Gene Expression Quantification Analysis

To reveal the dynamic expression changes in seed and SW tissues of the HOCL and LOCL during seed development, deep transcriptome sequencing was performed with the Illumina Hi-Seq^TM^ 4000 platform. RNA was extracted from the seed and SW tissue samples of both lines at 11, 16, 23, 30, 37, and 44 DAP. In total, 24 RNA libraries were subjected to paired-end sequencing separately, and 121 G of raw data were generated.

After quality control, 564,589,254 and 650,858,452 clean reads were obtained with an average of 47.0 million and 54.2 million reads per sample from the HOCL and LOCL, respectively. TopHat was used to map the reads to the *B. napus* genome, and detailed statistics are shown in Appendix A. On average, of the total clean reads from the HOCL and LOCL, 66.91% and 68.25% were mapped to unique localities of the genome, while 4.45% and 4.02% aligned to multiple locations, and the remaining 28.64% and 27.73% were unmatched (Appendix A).

When the fragments per kilobase of transcript per million mapped reads (FPKM) threshold was set to ≥1, there were 62,524 genes (61.9% of aligned genes) expressed in both the HOCL and LOCL, with 50,393 and 48,450 genes co-expressed in seeds and SW, respectively (Figure 2a–c). With FPKM values between 0.1 and 1, there were 12,030 and 11,811 expressed genes in the HOCL and LOCL, respectively, which were identified as weakly expressed partly due to many genes being tissue-specific and/or developmental stage-specific.

The numbers of expressed genes detected across the stages in both lines were quite even within the ranges of 37,202–40,566 and 36,837–41,993 in seeds and SW, respectively. Most of the expressed genes in the two tissues were co-expressed at a specific stage in both the HOCL and LOCL, while there were quite a few exclusively expressed genes (EG) at specific developmental stages in terms of such large quantities of expressed genes (Figure 2d); for example, only about 12.7% at the 11 DAP stage in seed and 12.4% in SW. Total expressed genes in seed and SW tissues of the HOCL and LOCL with locus details and FPMK values are shown in Appendix A.

The data of expressed genes in Figure 2 suggest that gene expression across all samples in seeds and SW of the HOCL and LOCL was uniform, which authenticated the fitness of RNA-Seq data and the experimental plan. The data also offer a valuable source for further identification of developmental stage-specific genes.

### 2.3. Expression Patterns of Genes Involved in Metabolic Pathways in High- and Low-Oil Content Lines

To understand the variations in expression patterns of genes associated with key metabolic pathways between the HOCL and LOCL, all EGs were classified by the MapMan metabolic pathway annotator. The 62,524 EGs in the HOCL and LOCL were classified into 35 metabolic functional categories (bins) (Table 1). Except for the category of no assigned genes (bin 35), there were 19 categories enriched with more than 300 EGs, indicating that gene transcription in associated metabolic pathways was active during seed development. We then compared relative expression between the HOCL and LOCL in each category at individual stages in seeds and SW. Overall, more genes were upregulated in the LOCL (30,313; Table 1) than in the HOCL (29,983). Notably, photosynthesis, lipid metabolism, development, and transport (metabolite transporters) were significantly enriched with more upregulated genes in the HOCL as compared to the LOCL based on Fisher’s exact test (Table 1).

Next, all EGs in seeds from 23 DAP were imported into the MapMan image annotator, version 3.6.0RC1, to assign them to associated pathways and subpathways (Figure 3). In lipid metabolism, photosynthesis, hormone metabolism, tetrapyrrole synthesis, development, and transport (metabolite transporters) enriched with more upregulated genes were preferentially expressed in the HOCL (shown in red) than the LOCL (shown in blue). Notably, in the lipid metabolism category, in subcategories related to lipid and TAG degradation, more genes were upregulated in the LOCL than in the HOCL. Similarly, some degradation-related subcategories in cell walls also showed more upregulated genes in the LOCL.

Stage-wise MapMan visualization of EGs presenting the overview of metabolism pathways dynamics in seed and SW tissues along with the development, giving the clear picture of the activity of specific pathway stage by stage and regulation of these pathways (Appendix A). For instance, Photosynthesis related bin was highly up-regulated in the HOCL at 16 and 23 DAP in both tissues and similarly lipid synthesis and transporter genes also show clear dynamics.

### 2.4. Comparative Dynamic Transcriptomics during Oil Accumulation between High- and Low-oil Lines

To get a thorough overview of transcriptomic dynamics related to the efficiency of oil accumulation in *B. napus*, differentially expressed genes (DEGs) were identified between the HOCL and LOCL in seed and SW tissues separately. In total, 3288 DEGs (Appendix A) were detected in seed and SW together, among which 1260 were found in both organs, 1041 in seed only, and 987 in SW only (Figure 4a).

Numbers of DEGs at different development stages varied within a relatively small range of 739 to 1168 in seed (Figure 4b). Stage-specific DEGs ranged from 107 to 200, with the least at 37 DAP (107) and the most at 44 DAP (200) (Figure 4b). There were 977 DEGs present at two or more time points, with 209 DEGs detected at all stages (Figure 4b).

Total DEGs at all developing stages of SW were identified ranging from 785 at 11 DAP to 1147 at 23 DAP (Figure 4c). The most abundant stage-specific DEG was at 11 DAP (259), and the least at 37 DAP (Figure 4c). There were 837 DEGs present at two or more stages, with 209 DEGs detected at all sampling time points (Figure 4c). Although the number of common DEGs present at all stages in seed and SW was the same, the genes were different.

To assess the trend of DEG expression pattern, up- and downregulated DEGs were quantified in all developmental stages. The overall number of downregulated DEGs was higher than the number of upregulated genes in both seed and SW tissues (Figure 5a). The change of upregulated DEGs across the developmental stages was fairly steady in both seed (383–431) and SW (305–473) compared with the downregulated DEGs in the two tissues (Figure 5b), which exhibited a sharp increase from 11 to 23 DAP, followed by a steadier change in the later stages (Figure 5b).

Furthermore, the seed and SW DEGs were subjected to Protein Analysis Through Evolutionary Relationships (PANTHER) for protein classification, and total protein class hits were 1104 and 1074 for in SW and seed, respectively. The most abundant class (about 47%) of DEGs was enzyme, followed by nucleic acid binding, miscellaneous proteins, transporters, and transcription factors in terms of abundance (Figure 5c).

### 2.5. qRT-PCR Analysis of Selected DEGs to Validate RNA-Seq Results

Quantitative real-time polymerase chain reaction (qRT-PCR) assay was performed to assess the reliability and validity of the RNA-Seq data with 20 randomly selected DEGs from seed and SW of both lines at 23 and 37 DAP. Gene copy–specific primers were used for all selected DEGs to verify the expression pattern by qRT-PCR and data were normalized by UBC-10 and actin genes independently (primer details are given in Appendix A). For direct comparison with RNA-Seq data, the relative expression quantified by qRT-PCR was transformed to log2 fold change (HOCL/LOCL). With both reference internal genes, the expression pattern of selected DEGs by RNA-Seq and qRT-PCR was highly correlated (*R* = 0.843–0.908 individually and overall correlation of 0.87 (Figure 6i)) at all selected developmental stages in both tissues (Figure 6), further confirming the reliability and consistency of our RNA sequencing data.

### 2.6. DEG Clustering

To better categorize the important co-expressing genes with similar expression dynamics, which possibly have similar regulation patterns in oil biosynthesis and accumulation, DEGs in seed and SW were clustered by Genesis (version 1.7.7) based on the K-means clustering method adjusted with normalized gene log2 fold change value (HOCL/LOCL) (Appendix A). Four distinct clusters in seed were observed (Figure 7a). DEGs in cluster 1 were upregulated from 11 to 30 DAP, with a peak at 23 DAP followed by a sharp downregulation (Figure 7a1); in cluster 2, genes in the HOCL were obviously upregulated until 44 DAP after a steady period at the first two stages (Figure 7a2). DEGs in cluster 3 were maintained at a low level with a marginal increase after downregulation at the first two stages (Figure 7a3), while DEGs in cluster 4 tended to be downregulated from 16 DAP to 37 DAP (Figure 7a4).

Similar to cluster 3 in seed, DEGs of cluster 1 in SW experienced a sharp downregulation at the first two stages and then turned to a steady level (Figure 7b1). Clusters 2 and 3 in SW exhibited similar patterns to clusters 1 and 2 in seed, respectively (Figure 7b). In SW cluster 4, DEGs were upregulated slightly at first and downregulated to the bottom at 30 DAP followed by a sharp upregulation (Figure 7b).

It is interesting to note that some clusters in seed and SW showed similar patterns, for example, seed cluster 2 versus SW cluster 3, and seed cluster 1 and SW cluster 2, while others between the two organs were different. To understand if the genes were similar or different, we carried out a Kyoto Encyclopedia of Genes and Genomes (KEGG) enrichment analysis for each cluster (Figure 7c,d). In SW cluster 2, DEGs included large numbers of genes in metabolism followed by carbohydrate metabolism, transporters, and transcription factors, while in SW cluster 3, metabolism and transporters were the two groups with the most abundant genes, indicating that an increase of gene expression in those categories for organs is important to achieve effective oil accumulation in the HOCL.

On the other hand, DEGs in seed cluster 1 were enriched in the pathways of metabolism, lipid metabolism, starch and sucrose metabolism, fatty acid biosynthesis, lipid biosynthesis protein, photosynthesis protein, and glycerol phospholipid metabolism. In SW cluster 2, metabolism, carbohydrate metabolism, and lipid metabolism were more abundant than other categories, while the most enriched genes were in the photosynthesis pathway (Figure 7).

For the clusters that exhibited different patterns in seed and SW, citrate cycle and starch and sucrose metabolism pathways were enriched in seed cluster 3, and metabolism, amino acid metabolism, carbohydrate metabolism, and lipid metabolism pathways were enriched in cluster 4, while SW clusters 1 and 4 did not have significant enrichment of any relevant pathway except the purine metabolism pathway in cluster 1 and metabolism pathway in cluster 4.

The above results clearly suggest that the pattern of important lipid and associated pathway genes are upregulated at the key developmental stages in the HOCL, while SW gene pathways provide valuable information to link SW (as a source) to seed (sink) to supply lipids or other assimilates for oil accumulation in seeds.

### 2.7. Functional Classifications of DEGs

To understand the functionality of the DEGs, Gene Ontology (GO) enrichment analysis was performed using Blast2GO [28] with a false discovery rate (FDR) cutoff value <0.05. Eventually, 2301 seed DEGs were categorized into 128 functional classes (terms). These classes were grouped into three major functional categories: Biological processes (BP, 84), molecular functions (MF, 20), and cellular components (CC, 24) (Figure 8a). Any term comprising more than 50 genes was regarded as significantly enriched. There were 48, 9, and 18 significantly enriched terms in the BP, MF, and CC categories, respectively (Appendix A). SW DEGs seemed more diverse, as 2247 DEGs were categorized into 287 functional classes, with 211 belonging to BP, 38 to MF, and 38 to CC (Figure 8c), of which 157, 8, and 37 terms were significantly enriched (Appendix A).

In the BP category of seed DEGs, the most abundant GO term was “response to stress” (28.4%), followed by “response to abiotic stimulus” (20.7%) and “small molecule metabolic process” (20.6%) (Figure 8b). In addition to the term “small molecule metabolic process,” the majority of other significantly enriched GO terms may be directly or indirectly involved in oil biosynthetic metabolism (23 terms altogether), such as “carbohydrate metabolic process” (12.5%), “lipid metabolic process” (11.5%), “carbohydrate derivative metabolic process” (10.9%), “cellular lipid metabolic process” (9.1%), and “lipid biosynthetic process” (8.3%) (Figure 8d).

On the other hand, the most abundant terms in the category BP of SW DEGs was “metabolic process” (67%), followed by “organic substance metabolic process” (57.3%), “cellular metabolic process” (54.7%), and “biosynthetic process” (37.7%) (Figure 8b). In addition, biological processes that may be directly or indirectly involved in photosynthesis, lipid biosynthesis metabolism, and transport were also significantly enriched, such as “carbohydrate metabolic process” (11.8%), “carbohydrate derivative metabolic process” (11.8%), “lipid metabolic process” (11.3%), “lipid biosynthesis” (8.4%), “photosynthesis” (3.1%), and “light reaction” (2.5%) (Figure 8e). It is interesting to note that hormone metabolism and regulation had a relatively high proportion in significantly enriched terms, such as “response to hormone” (15%), “regulation of hormone levels” (9%), “hormone metabolic process” (8%), and “hormone-mediated signaling pathway” (7.6%) (Figure 8e).

For the MF category in seed, the most enriched term was co-factor binding, followed by hydrolase activity, acting on glycosyl bonds along with hydrolase activity, and hydrolyzing *O*-glycosyl compounds, while in SW, most enriched terms were associated with oxidation/reduction, lyase, and transferase (data). In the CC category, the majority of seed DEGs were predicted to localize in “cytoplasm” and “cytoplasmic part” of the cell with other notable organelles, such as plastid, chloroplast, endoplasmic reticulum, endomembrane system, and vacuole, which are the key cellular sites for TAG biosynthesis. In SW, DEGs were predicted to localize in “cell” and “cell parts,” “cytoplasm” and “cytoplasm parts” of the cell with other notable organelles, such as plastid, chloroplast, photosynthetic membrane, and thylakoid (Appendix A), which are the key cellular sites for photosynthesis and TAG biosynthesis.

Upon GO analysis of stage-specific DEGs of seed and SW, we discovered that mostly carbohydrate and lipid metabolism-related GO terms were enriched in seed DEGs at 11, 23, and 30 DAP, while photosynthesis, carbohydrate, and lipid metabolism-related GO terms were enriched by SW DEGs at 11 and 16 DAP (Appendix A).

Taken together, the GO analysis thus provided important clues to identify genes involved in the formation of high oil content in seed and SW.

### 2.8. Identification of Up- and Down-regulated Pathways in Seed and SW

To assign the up- and downregulated DEGs to relevant metabolic pathways, the genes were subjected to KEGG analysis. The query datasets included 929 upregulated and 1375 downregulated seed DEGs, and 924 upregulated and 1326 downregulated SW DEGs. There were 435 upregulated DEGs in seed (seed-up, accounting for 46% of all upregulated DEGs in seed) and 410 seed-down (29%), and 401 SW-up (43%) and 418 SW-down (31%), which were annotated based on 24 KEGG metabolic pathways. The results are shown in Figure 9a,b, including 20, 12, 17, and 10 pathways corresponding to seed-up, seed-down, SW-up, and SW-down, respectively.

The above analysis revealed two types of metabolic pathways in seed and SW. The first included both up- and downregulated DEGs. For example, many up- and downregulated DEGs were grouped into “metabolism pathway” in seed or SW, and similar observations were also true for “carbohydrate metabolism pathway” in seed and “phenylpropanoid biosynthesis” in SW (Figure 9a,b). Such a pattern may indicate that different sets of DEGs are differentially regulated with increased or decreased expression in the HOCL to achieve a coordinated network for effective oil accumulation.

The second type contained only up- or downregulated DEGs. In seed, there were 12 pathways with only upregulated DEGs identified, and five pathways with only downregulated DEGs (Figure 9a,b). Notably, DEGs in pathways closely related to lipid metabolism were upregulated, such as fatty acid biosynthesis, sphingolipid metabolism, glycerolipid metabolism, lipid protein biosynthesis, and lipid metabolism, while DEGs in pathways involved in protein and carbohydrate accumulation were downregulated, such as starch and sucrose metabolism, amino acid metabolism, and synthesis. Such a pattern is consistent with the phenotype in the HOCL, in which lipid accumulation is more predominant than other storage, including proteins and carbohydrates. In SW, there were 11 pathways with only upregulated DEGs identified, and 6 pathways with only downregulated DEGs (Figure 9). DEGs involved in plant growth and photosynthate production in SW, such as the citrate cycle, circadian rhythm, and carbon fixation in photosynthetic organisms, were upregulated (Figure 9a,b). There were very many downregulated DEGs in SW assigned to several pathways, including biosynthesis of other secondary metabolites, protein families in signaling and cellular processes, exosome, biosynthesis of other secondary metabolites, carotenoid biosynthesis, alpha linolenic acid metabolism, and photosynthesis–antenna proteins in SW (Figure 9a,b). The exact implications of these observations are worth exploring.

## 3. Discussion

Oil is the predominant reserve and economic product in *B. napus* seeds. Understanding the dynamic regulation of seed oil content (SOC) during seed development is vital for precise breeding and genetic engineering of high oil variants in the crop, as even a slight percentage improvement in SOC in the field will lead to significant economic value [29]. In this study, temporally and spatially transcriptomic profiles were systematically compared across six developmental stages (11, 16, 23, 30, 37, and 44 DAP) in two tissues of seeds and SW from two inbred lines with highly stable and contrasting oil content. By integrating transcriptomic data with metabolite analyses, we identified critical expression patterns for effective oil accumulation and important genes in key pathways in seeds. Our work also provides global insights for understanding the role of SW in SOC biosynthesis and its regulation during the key developmental phase. To our knowledge, this is the first report of comparative transcriptomic profiling simultaneously with seed and SW tissues across intensive developmental stages in *B. napus*.

Our results suggest that the developmental phase from 11 to 44 DAP is the major oil biosynthesis stage, which is consistent with previous studies [23,30,31]. The time course and intervals adopted in this study seem to capture the dynamic changes in oil biosynthesis and related pathways [6]. Through comprehensive analysis, we identified 53,459 and 53,734 genes expressed in seeds and 51,767 and 51,463 in SW of the HOCL and LOCL, respectively. Similar percentages of expressed genes in seed and SW tissues were observed in Reference [32]. SW provides a link to the mother plant, through which nutrients and photosynthates are provided to seeds for oil biosynthesis [33,34]. Our gene expression data were generated by using seed and SW tissues in which the majority of EG in our dataset have a functional association with metabolic pathways along with regulatory pathways. For example, with DEG clustering, members of clusters 1 and 2 of seeds are mainly involved in metabolism, carbohydrate metabolism, lipid metabolism, sucrose and starch metabolism, transporters, transcription factors, FA biosynthesis, lipid biosynthesis protein and photosynthesis protein, and glycerophospholipid metabolism. Cluster 4 also includes 17 genes in the lipid metabolism pathway, one of which is the FA and TAG degradation gene.

Genetic interactions play an important role in biological functions [35]. GO analysis of seed SW DEGs shows that the majority of genes are enriched in metabolic pathways and biosynthesis. Interestingly, a large fraction of genes involved in response to stimulus and stress pathways is also observed. The reason for this could be, first, that lipid-related genes are often participants responding to stress [36,37,38,39], and second, that there are indeed stresses occurring during a relatively long growth period of seed development and these genes may function as a stabilizer for metabolism.

After embryogenesis, carbohydrates are mainly utilized in lipid biosynthesis [12], and glucose, sucrose, and starch are the major suppliers of carbon flux during oil biosynthesis [40,41]. In our results it is noted that, except for a few stages, glucose, sucrose, and starch contents were significantly higher in the HOCL. The sucrose content curve for seed was intersected just after 30 DAP, which shifted the trend to LCOL (Figure 1B). This may be due to more consumption of sucrose in the HOCL to produce more oil during the major oil biosynthesis phase [12]. The timing of carbohydrates is slightly ahead of oil biosynthesis.

Phytohormones play a very important role in FA biosynthesis [11]. Several GO terms related to hormones are found significantly enriched in SW DEGs. Auxin, abscisic acid (ABA), and jasmonate (JA) are most affected hormones. JA content in seeds is significantly higher in the LOCL at 16, 23, and 30 DAP, and significantly higher in the HOCL at 37 and 44 DAP (Figure 10a). On the other hand, JA content in SW is significantly higher at 16 DAP in the HOCL (Figure 10d). JA is important for biological processes like growth and development of the plant. JA is synthesized from the acyl-CoA pool produced from the FA biosynthesis pathway [42], and alteration in FA production has been observed in JA and auxin-deficient A. thaliana mutant [11]. On the contrary, JA signaling has been compromised by mutations in key FA biosynthetic genes in Arabidopsis ssi2 locus [43,44]. This phenomenon may indicate the presence of a regulatory loop among the FA biosynthesis and JA signaling pathways. It was reported previously that ABA has a close relationship with FA biosynthesis and transportation [45]. Application of ABA on *B. napus* rapidly induces oleosine and FA elongase gene expression and enhances TAG accumulation [46]. ABA content is significantly high in the HOCL at 30, 37, and 44 DAP in seed and at 16 and 44 DAP in SW (Figure 10). Auxin plays a very important role in plant development and FA biosynthesis [11]. Auxin content is also significantly high in the HOCL at 16, 23, and 44 DAP in seed, and at 23, 30, and 37 DAP in SW. There are 18, 24, and 15 differentially expressed genes among seed and SW tissues of the HOLC and LOCL that respond to JA, ABA, and auxin, respectively (Appendix A). Auxin genes show similar expression to FA genes. Expression dynamics of differentially expressed JA, ABA, and auxin genes in seed and SW tissues of the HOCL vs. LOCL are shown in a heat map in Figure 11. The inconsistency of hormone content and expression of differentially expressed genes in seed and SW showed tissue-specific regulation of metabolite and transportation.

Furthermore, 75 acyl lipid metabolism genes (Appendix A), and 42 transcription factor genes (Appendix A) were differentially expressed in the two lines, including a number of renown oil synthesis and regulatory genes, for example, genes for Diacylglycerol Acyltransferase and Monoacylglycerol Acyltransferase (*DGAT*, *MAGAT*), Ketoacyl-ACP Synthase 1 (*KAS1*), Ketoacyl-CoA Synthase (*KCS1*), nuclear factor Y, subunit B (*NF-YB*), *LEAFY COTYLEDON 1* (*LEC1*) and *LEC1-LIKE*, which are directly and indirectly involved in carbohydrate and lipid metabolism and enhance the seed oil production in rapeseed [47] and maize [48]. Uncover of these genes, together with the ones involved in hormone regulation (Appendix A and Figure 11), provided a comprehensive understanding of gene expression pattern in high oil content line. These genes can be the targets for further molecular characterization.

## 4. Materials and Methods

### 4.1. Plant Material

Two semiwinter-type inbred lines of *B. napus*, 1L99 (high-oil-content line, HOCL) and 1L363 (low-oil-content line, LOCL) with significantly different oil content were grown in experimental fields in Wuhan and Chongqing, China, for 2 consecutive seasons in 2013–2014 and 2014–2015. The HOCL accumulated ~13% more oil content than the LOCL. These 2 inbred lines were then planted during the 2016–2017 season in the experimental field of Huazhong Agricultural University, Wuhan. At the time of flower initiation, controlled artificial self-pollination was executed with individual flowers to ensure that silique development could be dated exactly. Seed and silique wall (SW) tissues were sampled with 6 biological replicates at 11, 16, 23, 30, 37, and 44 days after pollination (DAP). At each stage, seeds were separated from silique wall gently on ice as previously described in Reference [11] and immediately frozen in liquid nitrogen and stored at −80 °C until total RNA extraction, fatty acid analysis, and carbohydrate quantification.

### 4.2. RNA Extraction, cDNA Library Construction, and Sequencing

To extract total RNA for sequencing, tissues from 3 biological replicates from each stage were pooled and milled to a fine powder and used for isolation of RNA by using a Plant Total RNA Extraction Kit (TransZol Plant, BioTeke, Beijing, China) following the manufacturer’s instructions. The RNA was subjected to RNase-free DNase-I (Thermo Scientific, Waltham, MA, USA) treatment to remove gDNA contamination and then quantified with nucleic acid/protein analyzer DU^®^730 (Beckman Coulter^®^, Brea, CA, USA). The RNA integrity was assessed by the Experion™ automated electrophoresis system (Bio-Rad Laboratories Inc., CA, USA). Samples with an RNA quality indicator (RQI) of more than 7 were considered acceptable for library construction. All 24 samples, including seed and SW tissues collected from the 6 stages of both lines, were subjected to library construction by using the TruSeq™ Illumina^®^ (San Diego, CA, USA) RNA Sample Preparation Kit following the manufacturer’s protocol. RNA of good quality and acceptable integrity was further processed for poly-A-containing mRNA cleansing, fragmentation of mRNA, synthesis of double-stranded cDNA, and amplification through polymerase chain reaction (PCR). Sequencing of cDNA was performed on an Illumina HiSeq^TM^ 4000 platform. Paired-end reads with a length of 2 × 150 base pairs (bp) were generated.

### 4.3. Quality Control and Read Mapping to Reference Genome

The raw data of the Illumina^®^ sequence reads in fastq format were subjected to a filtration process using the NGS-QC Toolkit v2.3 software package [49]. The process includes (1) removing the reads with primer/adaptor sequences and substandard reads with a sequence quality score (Q-score) beyond the standard 30%; (2) trimming the first 10 uncertain base pairs of reads that were determined by the percentages of 4 nucleotides (A, T, C, and G) and low-quality bases (Q-score < 20); and (3) discarding reads less than 50 base pairs in length. Finally, quality control of preprocessed reads was evaluated by FastQC v_0.11.1 (Babraham Bioinformatics, Cambridge, UK). Subsequently, TopHat v2.0.11 [50] was employed with default parameters for mapping of the high-quality reads to the *B. napus* genome [20]. Reference genome and corresponding gene annotation files were directly retrieved from the Genoscope (http://www.genoscope.cns.fr/brassicanapus/data/).

### 4.4. Differential Gene Expression Quantification

Identification of differentially expressed genes (DEGs) and gene expression level assay were carried out with Cufflinks v2.2.0 [51] by using the uniquely mapped reads. Fragments per kilobase of transcript per million mapped reads (FPKM) was calculated as a measure of the level of gene expression. DEGs among all developmental stages of seed and SW samples of the HOCL and LOCL were recognized by Cuffdiff [52] with a cutoff of log2 fold changes ≥ 1 and a false discovery rate (FDR) ≤ 0.05.

### 4.5. MapMan Visualization and Enrichment Analysis

Expressed genes with FPKM cutoff > 1 in the HOCL and LOCL were articulated as relative expression ratio (log2 fold change) of the HOCL relative to LOCL, and then subjected to MapMan version 3.6.0 RC1 image annotator (http://mapman.gabipd.org/web/guest/mapman) [53] with *Brassica_napus* annotation_v5 mapping reference data to group and display them graphically in metabolic pathways. The Wilcoxon rank sum test in the MapMan program was used to determine which bins or sub-bins were differentially regulated between the 2 genotypes.

### 4.6. Gene Ontology and Enrichment Analysis and KEGG Pathway Detection

For gene function annotation, all 101,040 *B. napus* genes were subjected to search against the nonredundant protein database (Nr) of the National Center for Biotechnology Information using BlastP with E-value ≤ 1 × 10^−5^. Gene Ontology (GO) terms associated with every BLAST hit were annotated using the Blast2GO tool (version 4.0.7) [28]. Then, all *B. napus* genes were subjected to search by the InterPro database using the InterProScan-5 tool (https://www.ebi.ac.uk/interpro/interproscan.html) [54]. Finally, the *B. napus* gene GO terms were annotated by combining the annotation outcomes from Blast2GO and InterPro. Gene ontology enrichment analysis was performed by employing the Blast2GO program with FDR ≤ 0.01. All GO terms that were significantly enriched in differentially expressed genes as compared to genome background were displayed. The redundancy of significantly enriched GO terms was reduced using REViGO with a similar cutoff of 0.75 [55]. Furthermore, homologs of A. thaliana genes in the *B. napus* genome were identified by the BlastP online program with E-value ≤ 1 × 10^−5^, coverage ≥ 50%, and identity ≥ 50%. In order to further validate the Blast2GO annotation, KEGG classification of up- and downregulated DEGs in seed and SW was performed separately by using the BlastKOALA tool with *p* < 0.05 [56]. Additionally, DEGs were categorized by protein classes by subjecting them to analysis in the PANTHER classification system [57].

### 4.7. Oil Content, Carbohydrates, and Hormone Quantification

All developmental stages of seed and silique wall tissues, the same as for RNA-Seq, were used for oil content, carbohydrate, and hormone quantification. Gas chromatography with flame ionization detector (GC-FID) was used for oil content quantification according to the protocol as described in Reference [58] with minor modifications. GC was programmed as described in Reference [58]. Fatty acids were identified and determined by comparing retention times to standards (C17:0).

Soluble sugar, sucrose, and starch content of the sampled seed and silique wall were extracted by the phenol sulfuric acid method [59,60] using an Infinite M200 PRO spectrometer (Tecan^®^, Grodig, Austria).

Plant hormone jasmonic acid (JA), abscisic acid (ABA), and indole acetic acid (IAA) levels in developing seed and SW were estimated by employing high-performance liquid chromatography. Samples were prepared by using a crude extraction procedure originally reported in Reference [61] with slight modifications. In triplicate, 100 mg of each ground sample was homogenized with 1 mL 80% methanol (HPLC grade) supplemented with internal standard and briskly shaken at a frequency of 300 rpm at 4 °C overnight. The subsequent extraction process was performed as described in Reference [62]. After centrifugation at 13,000 rpm for 15 min at 4 °C, the supernatant was filtered and dried by evaporation under the flow of nitrogen gas for approximately 4 h at room temperature. Finally, the extract was dissolved in 200 μL of 50% methanol and hormones were quantified by HPLC.

### 4.8. Validation of RNA-Seq Data by qRT-PCR Analysis

Quantitative real-time polymerase chain reaction (qRT-PCR) analysis was performed to validate the expression patterns of genes obtained by Illumina^®^ sequencing [63]. Expression profiles of 20 randomly selected DEGs were studied in seed and SW tissues at 23 and 37 DAP, the same tissue samples that were used for RNA sequencing. Gene-specific primers for qRT-PCR were designed for the selected genes/transcripts from the transcriptome data by using the Primer Premier-5 program. The primer details of all genes under study are listed in Appendix A. The qRT-PCR assay was carried out as described in Reference [11] with biological triplicates repeated twice (technical replicates) for each reaction [64]. Transcript level was measured by the comparative cycle threshold (*C*_t_) method. UBC-10 (BnaA10g06670D) [1] and BnaActin (GenBank accession number AF111812) [11] were used as references to normalize the expression data. Quantitative RT-PCR analysis was performed with SYBR-Green MasterMix. The relative cycle threshold (*C*_t_) was used to calculate relative expression by the 2^−ΔCt^ method. Comparative expression in 1L99 and 1L363 finally transformed to log2 fold changes [65].

## Figures and Tables

**Figure 1 ijms-20-01982-f001:**
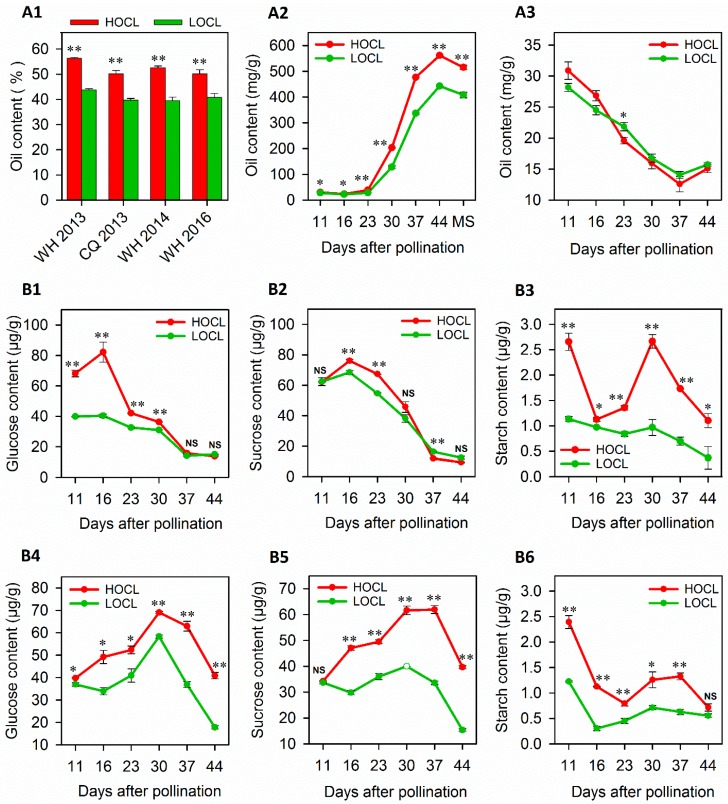
Phenotypic variation of the high-oil-content line (HOCL) and low-oil-content line (LOCL) across all developmental stages. (**A**) Dynamics of oil accumulation in seeds and silique wall of the HOCL and LOCL. (**A1**) The oil content of two lines in Wuhan (WH) and Chongqing (CQ) in two growth seasons (2013–2016). (**A2**) Dynamics of oil content in developing seeds of the HOCL and LOCL at six developmental stages, including mature seed (MS). (**A3**) Dynamics of oil content in developing silique wall of the HOCL and LOCL at six developmental stages. (**B**) Dynamics of carbohydrates in seeds and silique wall tissues of the HOCL and LOCL at all six selected developmental stages in Wuhan during the 2016–2017 growth season. (**B1**) Glucose content in seed. (**B2**) Sucrose content in seed. (**B3**) Starch content in seed. (**B4**) Glucose content in silique wall (SW). (**B5**) Sucrose content in SW. (**B6**) Starch content in SW. Error bars represent standard error (*n* = 3). * and ** indicate significant difference and NS indicates no significance between the HOCL and LOCL at *p* < 0.05 and *p* < 0.01 levels, respectively (one-tailed *t*-test).

**Figure 2 ijms-20-01982-f002:**
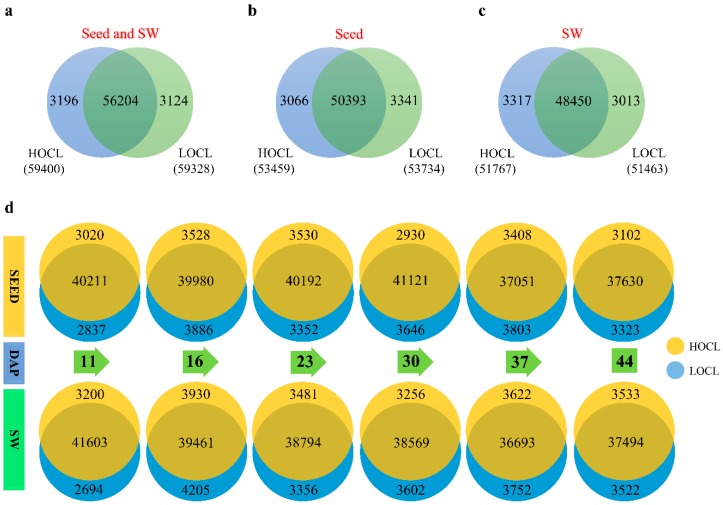
Genes expressed (fragments per kilogram of transcript per million mapped reads (FPKM) ≥ 1) in seed and silique wall of the HOCL and LOCL. (**a**) Seed and SW; There were 59,400 and 59,328 genes expressed in seed and silique wall of the HOCL and LOCL, respectively, with 56,204 genes co-expressed in both lines and 3196 and 3124 specifically expressed in each line. (**b**) Seed; There were 50,393 co-expressed genes, 3066 (HOCL) and 3341 (LOCL) genotype-specific expressed genes, and 53,459 and 53,734 total genes expressed in the HOCL and LOCL, respectively (**c**) Silique wall; There were 48,450 co-expressed genes, 3317 (HOCL) and 3013 (LOCL) genotype-specific genes, and 51,767 and 51,463 total genes expressed in the HOCL and LOCL, respectively. (**d**) Dynamics of genes expressed in seed and SW tissues of the HOCL and LOCL along with the development, and changing expression patterns of genes along with the development of seed and SW of the HOCL and LOCL.

**Figure 3 ijms-20-01982-f003:**
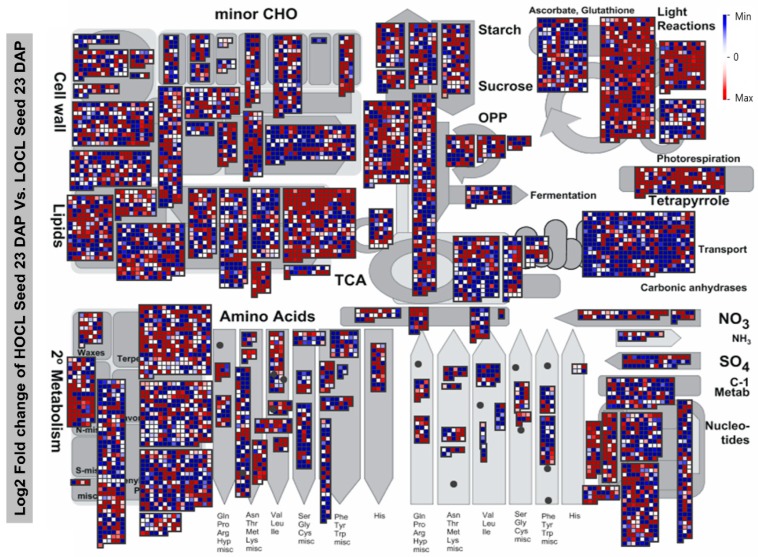
Expressed genes in seed tissue at 23 DAP involved in major pathways revealed by MapMan. A total of 7175 genes are visible in data points. Red dots represent genes expressed preferentially in the HOCL, and blue, those expressed in the LOCL.

**Figure 4 ijms-20-01982-f004:**
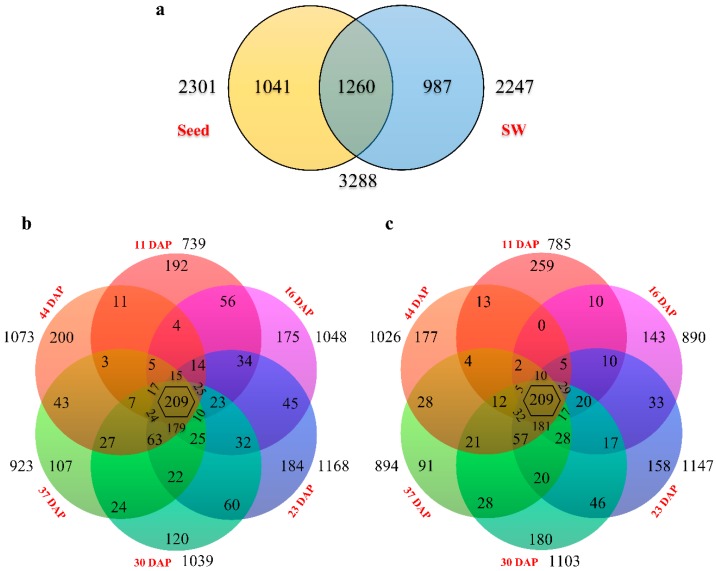
Overview of differentially expressed genes (DEGs) in seed and silique wall of the HOCL compared to LOCL (log2 fold changes ≥ 1 and false discovery rate (FDR) ≤ 0.05). Uniquely mapped reads were used to calculate overlapping regions corresponding to the number of DEGs present at more than one developmental stage. (**a**) DEGs expressed in seed and silique wall of the HOCL as compared to LOCL. (**b**) DEGs expressed at six developmental stages of seeds were broken down into DEGs that were unique at one stage or overlapped in 2, 3, 4, or 5 stages or co-expressed at all stages under study. (**c**) DEGs expressed at all six developmental stages of SW were broken down into DEGs that were unique at one stage or overlapped in 2, 3, 4, or 5 stages or co-expressed at all stages under study.

**Figure 5 ijms-20-01982-f005:**
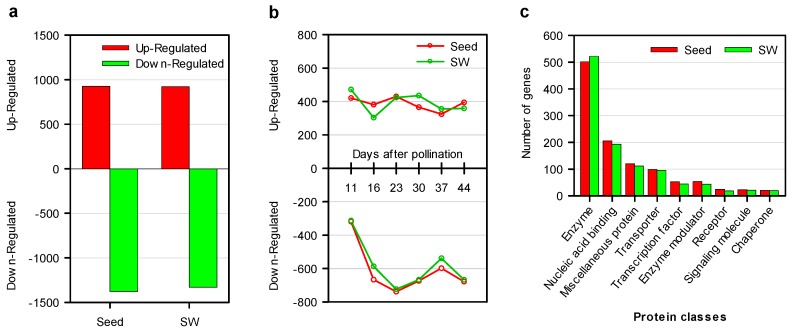
Up- and downregulated DEGs in seed and silique wall at different developmental stages. (**a**) Up- and downregulated DEGs of the HOCL as compared to LOCL. (**b**) Overall up- and downregulated DEGs in seed and SW of the HOCL. (**c**) Protein classification of DEGs using Protein Analysis Through Evolutionary Relationships (PANTHER). Total protein class hits were 1104 and 1074 in SW and seed, respectively.

**Figure 6 ijms-20-01982-f006:**
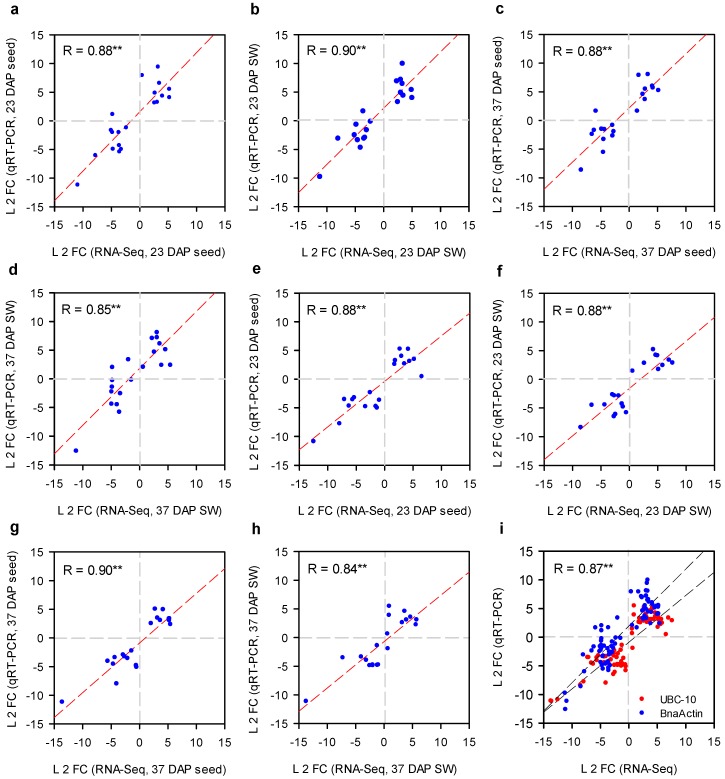
Validation of RNA sequencing data by using quantitative real-time polymerase chain reaction (qRT-PCR). Expression abundance of 20 randomly selected differentially expressed genes in seed and silique wall samples 23 and 37 days after pollination is presented as the ratio of HOC and LOC lines. *B. napus* actin and UBC-10 genes were used as an internal control for data normalization. Normalized expression (2^−ΔCt^) in 1L99 samples was divided by normalized expression in 1L363 samples and log2 transformed. Data represent mean values ± SE of three biological replicates and two technical replicates of each sample. The correlation coefficient (R) was calculated for comparison. BnaActin: (**a**) 23 DAP seed, (**b**) 23 DAP SW, (**c**) 37 DAP seed, (**d**) 37 DAP SW, UBC-10, (**e**) 23 DAP seed, (**f**) 23 DAP SW, (**g**) 37 DAP seed, (**h**) 37 DAP SW, (**i**) collective correlation coefficient of both stages and tissues together. ** Significant at *p* < 0.01. L 2 FC, log2 fold change.

**Figure 7 ijms-20-01982-f007:**
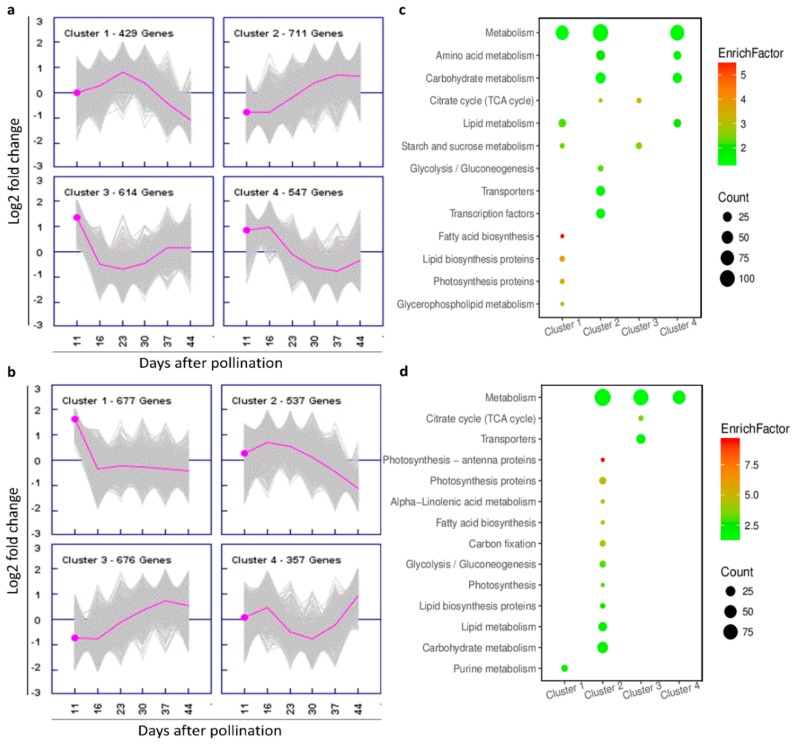
Cluster analysis of seed and SW DEGs. Four clusters in each tissue were identified by Genesis based on K-means clustering adjusted with normalized gene log2 fold change value. The y-axis represents normalized gene log2 fold change value. (**a**) DEG cluster expression in the seed. (**b**) DEG cluster expression in silique wall. (**c**) Kyoto Encyclopedia of Genes and Genomes (KEGG) analysis of genes in seed clusters. (**d**) KEGG analysis of genes in SW clusters (only highly enriched, most relevant KEGG pathways are shown here). (Gene number ≥ 5 and *p*-value ≤ 0.05.) Enrichfactor refers to the ratio of differential genes in a cohort group compared with the ratio in the database.

**Figure 8 ijms-20-01982-f008:**
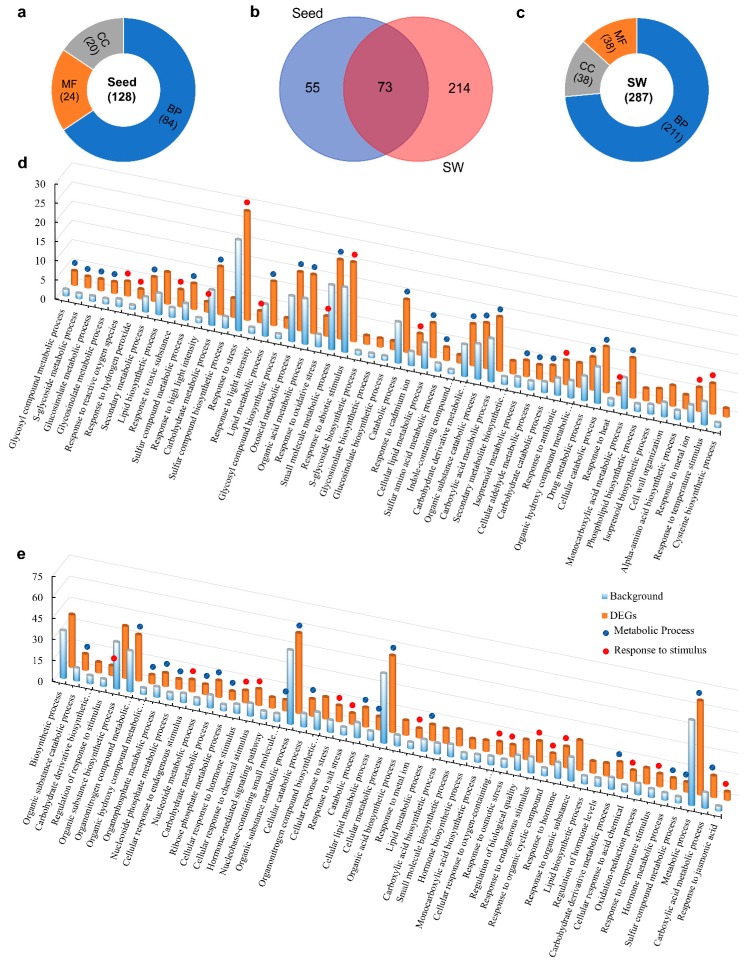
Gene Ontology (GO) enrichment analysis of seed and SW DEGs. (**a**) A number of GO terms in each category enriched by seed DEGs. (**b**) GO terms uniquely enriched by seed or SW DEGs and commonly enriched terms by both tissues. (**c**) A number of GO terms in each category (BP, MF, and CC) enriched by seed DEGs. (**d**) Top 50 GO terms included in the biological process category in seed DEGs (**e**) Top 50 GO terms included in the biological process category in SW DEGs. The y-axis is the percentage of genes mapped by the term, representing the abundance of the GO term. Percentages for the input list were calculated by the number of genes mapped to the GO term divided by the number of all genes in the input list; the same calculation was applied to the reference list (background). BP, biological process; MF, molecular function; CC, cellular component.

**Figure 9 ijms-20-01982-f009:**
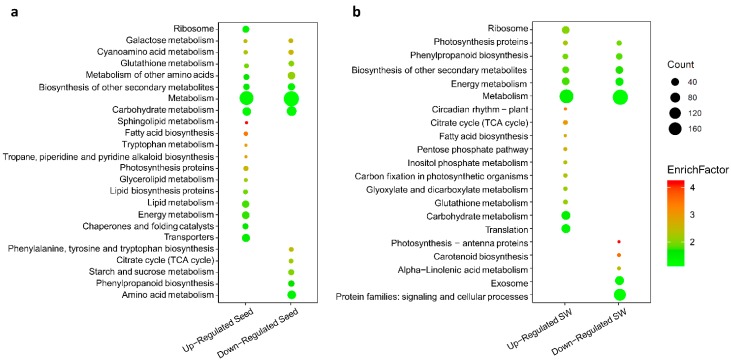
Identification of pathways enriched by up- and downregulated DEGs at all developmental stages in (**a**) seed and (**b**) SW of the HOCL and LOCL by KEGG analysis (gene number ≥ 5 and *p*-value ≤ 0.05). Enrichfactor refers to the ratio of differential genes in a cohort group compared with the ratio in the database.

**Figure 10 ijms-20-01982-f010:**
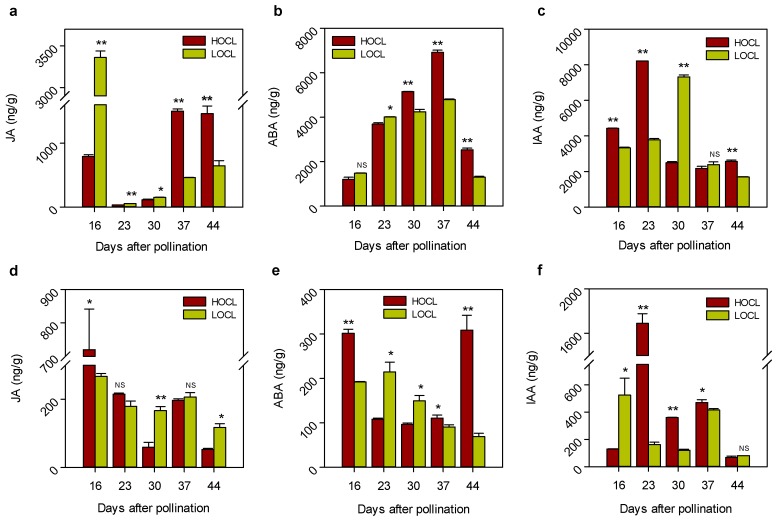
Comparison of hormone content in seed and SW between the HOCL and LOCL during seed development. (**a**–**c**) Jasmonic acid, abscisic acid, and indole acetic acid in seed. (**d**–**f**) Jasmonic acid, abscisic acid, and indole acetic acid in SW. Error bars represent standard error (*n* = 3). * and ** indicate significant difference and NS indicates no significance between the HOCL and LOCL at *p* < 0.05 and *p* < 0.01 levels, respectively (one-tailed *t*-test).

**Figure 11 ijms-20-01982-f011:**
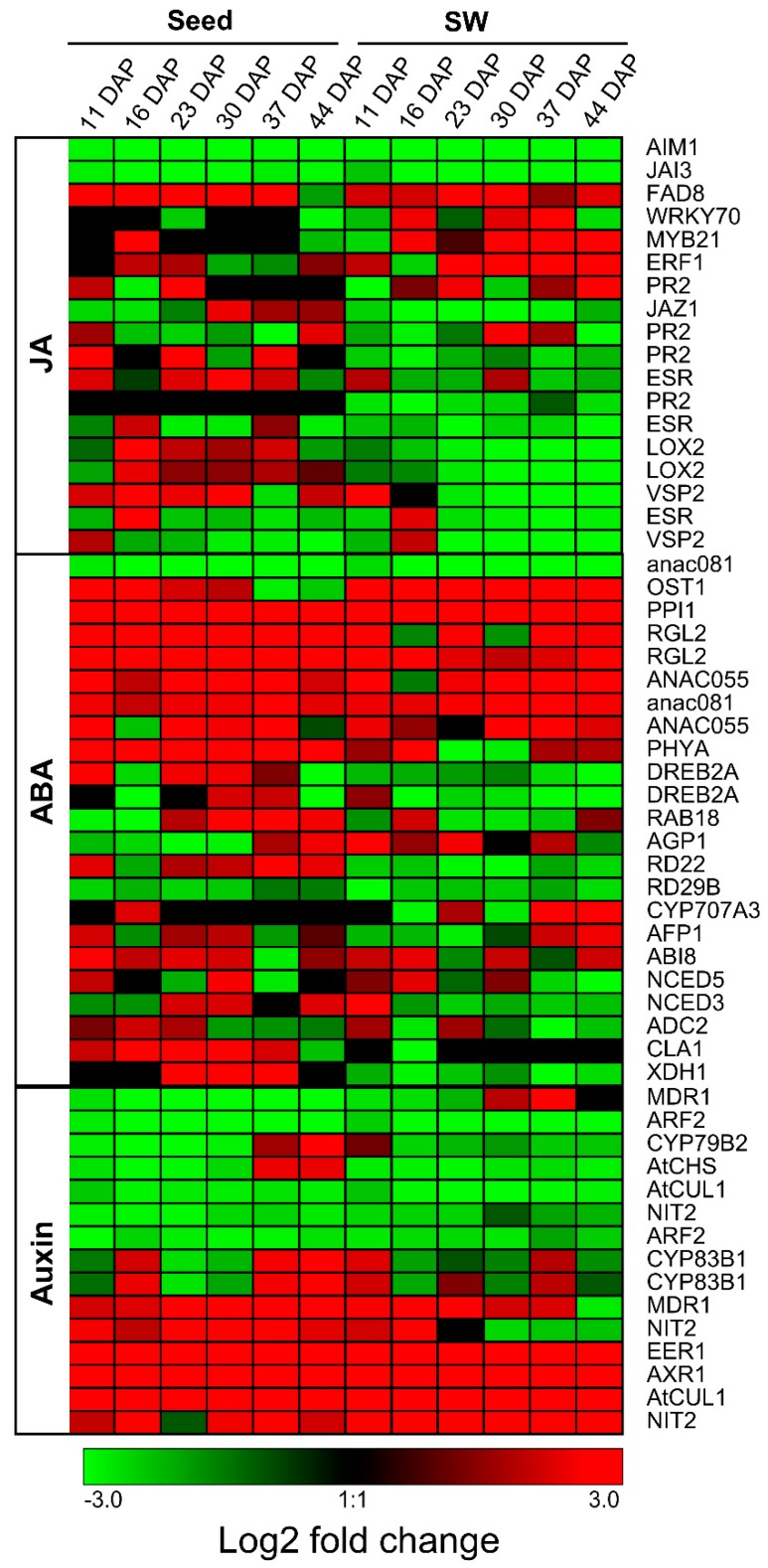
Heat map of differentially expressed jasmonate (JA), abscisic acid (ABA), and auxin genes. Red and green represent upregulated genes in the HOCL and LOCL, respectively.

**Table 1 ijms-20-01982-t001:** Overview of gene expression related to metabolic pathways in the HOCL and LOCL during seed-oil accumulation at 23 days after pollination (DAP).

Bin	Assignment ^a^	All ^b^	HOCL-Up	LOCL-Up	*p*-Value
1	Photosynthesis	556 (0.89)	353 (1.18)	193 (0.64)	2.51 × 10^−13^ **
2	Major carbohydrates metabolism	268 (0.43)	127 (0.42)	134 (0.44)	0.77
3	Minor carbohydrates metabolism	318 (0.51)	150 (0.50)	158 (0.52)	0.60
4	Glycolysis	233 (0.37)	103 (0.34)	125 (0.41)	0.07
5	Fermentation	42 (0.07)	14 (0.05)	28 (0.09)	1.8 × 10^−3^ **
6	Gluconeogenesis/glyoxylate cycle	44 (0.07)	19 (0.06)	20 (0.07)	0.76
7	Oxidative PP cycle	86 (0.14)	48 (0.16)	38 (0.13)	0.35
8	TCA/organic acid transformation	235 (0.38)	83 (0.28)	138 (0.46)	1.0 × 10^−3^ **
9	Mitochondrial transport/ATP synthesis	366 (0.59)	103 (0.34)	254 (0.84)	6.0 × 10^−16^ **
10	Cell wall (Degradation)	1178 (1.88)	511 (1.70)	574 (1.89)	0.02 *
11	Lipid metabolism (FA desaturation)	1133 (1.81)	556 (1.85)	524 (1.73)	4.6 × 10^−2^ *
12	N-metabolism	80 (0.13)	40 (0.13)	40 (0.13)	0.81
13	Amino acid metabolism	800 (1.28)	345 (1.15)	432 (1.43)	2.7 × 10^−3^ **
14	S-assimilation	42 (0.07)	16 (0.05)	25 (0.08)	0.18
15	Metal handling	187 (0.30)	77 (0.26)	97 (0.32)	0.17
16	Secondary metabolism	1122 (1.79)	469 (1.56)	529 (1.75)	0.06
17	Hormone metabolism	1524 (2.44)	671 (2.24)	657 (2.17)	0.25
18	Co-factor and vitamin metabolism	216 (0.35)	87 (0.29)	126 (0.42)	0.01*
19	Tetrapyrrole synthesis	122 (0.20)	79 (0.26)	39 (0.13)	9.2 × 10^−6^ **
20	Stress (biotic and abiotic)	2379 (3.80)	978 (3.26)	1114 (3.67)	8.6 × 10^−4^ **
21	Redox regulation (Thioredoxin)	604 (0.97)	227 (0.76)	329 (1.09)	4.18 × 10^−5^ **
22	Polyamine metabolism	55 (0.09)	30 (0.10)	23 (0.08)	0.06
23	Nucleotide metabolism	482 (0.77)	221 (0.74)	248 (0.82)	0.14
24	Biodegradation of Xenobiotic	78 (0.12)	26 (0.09)	49 (0.16)	1.6 × 10^−3^ **
25	C1-metabolism	101 (0.16)	35 (0.12)	63 (0.21)	5.8 × 10^−3^ **
26	Miscellaneous enzyme families	3319 (5.31)	1431 (4.77)	1520 (5.01)	0.20
27	RNA	7875 (12.6)	3950 (13.1)	3434 (11.3)	2.4 × 10^−12^ **
28	DNA (Synthesis repair)	1127 (1.80	641 (2.14)	466 (1.54)	1.4 × 10^−9^ **
29	Protein	9862 (15.7)	4909 (16.3)	4652 (15.3)	4.6 × 10^−3^ **
30	Signaling	3222 (5.15)	1508 (5.03)	1470 (4.85)	0.94
31	Cell (Vesicle transport)	2150 (3.44)	1036 (3.46)	1073 (3.54)	0.92
32	Micro RNA, natural antisense etc.	3 (0.004)	0 (0)	3 (0.01)	0.47
33	Development	2027 (3.24)	958 (3.20)	885 (2.92)	1.9 × 10^−2^ *
34	Transport (Metabolite transporters)	2670 (4.27)	1306 (4.36)	1235 (4.07)	5.1 × 10^−7^ **
35	Not assigned	20,586 (32.92)	8876 (29.60)	9618 (31.7)	4.8 × 10^−8^ **
**Total**		**62,524**	**29,983**	**30,313**	

(a) All expressed genes were classified into functional categories using MapMan 3.6.0 RC1. (b) The number of expressed genes in each functional category was compared to the total number of expressed genes within a similar functional group. Numbers in parentheses indicate the percentage of the total number of expressed genes within a functional group. *p*-values were calculated using Fisher’s exact test. Functional categories significantly enriched or reduced represent a given number of genes relative to the total number of genes. * Significant difference at the 0.05 probability level; ** significant difference at the 0.01 probability level.

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
