# Peer review of "Comparative Transcriptome Analysis of Developing Seeds and Silique Wall Reveals Dynamic Transcription Networks for Effective Oil Production in Brassica napus L."

_ijms, 2019, doi:10.3390/ijms20081982_

Reviewer 1 Report

This manuscript describes the transcriptomic analysis of developing seeds and silique wall in two contrasting inbred lines of Brassica Napus  with ~13% seed oil content difference. In theory, this work could be useful to understand the molecular networks that are involved in oil production during Brassica Napus development. Unfortunately, in my opinion, the paper presents several issues that need to be solved for a possible publication.

1)      the English language and style require extensive editing; the main text presents a number of  spelling and grammar errors; moreover the meaning of some sentences results difficult to understand. Here some examples:

line 53: “and provide the source” instead of “provides”;

line 54: “and provide the nutrients” instead of “provides”;

line 192: “some degradation related subcategories in cell wall are also show” instead of “shown”;

line 273: “with a margin increase” instead of “marginal”;

line 304: “ SW cluster 1 and 4 has have not enriched “;

line 229-232: the sentence is not so clear.

2)      Although the statistical and bioinformatics analysis are methodologically correct and complete, data are not well described and, in some cases, the figures are not clear:

Figure 2a-b-c: the interpretation of Venn diagrams is quite difficult, because they are not named (figure 2a-Seed and SW; figure 2b-Seed; figure 2c-SW).

Figure 4: It is not clear because there are too many numbers and it should be qualitatively improved.

3)      The manuscript doesn’t describe in detail  the candidate genes for oil biosynthesis and their trends in the two lines during the development. I think this should be the main goal of this work; it is evident a massive experimental and bioinformatics work, but the paper lacks of biological interpretation of the results, especially about the genes directly involved in oil production.  In line 407 the authors wrote “by the integration of transcriptomic data with metabolites analyses, we identified critical expression pattern for effective oil accumulation and important genes in key pathways in seeds”, but I didn’t find a list of such genes and their expression pattern in the main text or figures. It could be important to analyse the expression trends of candidate genes by qRT-PCR and to compare them with transcriptomic data.

Author Response

Response to comment 1).

     We greatly appreciate the reviewer’s careful reading of our manuscript. The manuscript was further language edited by MDPI professional English editing service (Certificate is attached). We also further improved the writing throughout the manuscript based on the editing. 

Response to comment 2).

     Figure 2a,b,c; We have modified the figures legends and description in relevant figures as suggested. Examples; Figure 2 legend, line 162, 164, 166.

     Figure 4;  We understand the concern. The figure includes all differentially expressed genes (DEGs) at six sampling stages and overlapped DEGs at 2, 3, 4, 5 or all stages as well, which was aimed to reveal the dynamic changes of DEG numbers, especially the ones common at two or more stages.  Indeed, we compared different presentation formats for such multiple dimensional data and found the current way is easier to follow, though the numbers on Figure 4b and 4c are somehow crowded. To further improve the visibility and quality of the figure, we increased the size and resolution of the figure

Response to comment 3).

     We appreciate the insightful comments. To address the concern, we further supplied a list of DEGs involved in acyl lipid metabolism (Supplementary Table 3), including a number of known genes for oil synthesis, in the revised version. Furthermore, we added new data of transcription factors differentially expressed in the two lines (Supplementary Table 4). These genes, together with the ones involved in hormone regulation (Supplementary Table 2 and Figure 10) provide a comprehensive understanding of gene expression pattern in high oil content line and thus can be the targets for further molecular characterization of individual genes (Line 479-488).

To verify the RNA-Seq data, we did conduct qRT-PCR with a set of genes and the expression trends are coincident with transcriptomic data (Figure 6). Therefore, the conclusions based on the transcriptomic data are reliable and more comprehensive. It is also understandable that it will be not practical to run qRT-PCR to verify genes in different pathways with such a large volume of transcriptomic data. 

Reviewer 2 Report

The manuscript by Shahid et al. performed the transcriptome analysis in the developing seeds and silique wall (SW) from two inbred lines with 13% seed oil content difference. Authors also measured seed oil content, carbohydrates content, and accumulation of hormones in developing seeds and SW. The current manuscript is of significant interest to the agricultural and industrial community. Overall, the manuscript (MS) is well written, and conclusions are well supported by the data presented. Experimental design, samples collection and statistical analysis were performed with great care.

However,  I still believe that MS could be improved significantly:

Major concerns:

Figure 1: Authors did not interpret the results for B1-B6.  Interpretation of the differences observed for carbohydrate contents required and what is the significance of this to seed oil content?. Also, it is not clear whether the data presented for developing tissues is from one location/growth season?

Include this information in the figure legend. 

Table1: It is not clear from the MS or the table that genes considered for the functional categories are from the particular developmental stage or the total genes from all developmental stages.  Please state it explicitly in the running MS and figure legends of all the analysis did in the entire MS.  It could be more informative if the authors do these analyses in two sections: one is at the developmental stages where oil content between two lines shows no difference (DAP11-23) and another at the stage where seed oil content differ between two lines (DAP30-44).

Figure 6: It is surprising that why authors picked genes randomly for the qRT PCR analysis. It could be more relevant to the study if authors pick genes highly conserved for oil biosynthesis and interpret those results in terms of oil content differences observed in two lines. 

  Figure 11. Explain methods for hormone measurement in a bit more details in the method section. Extracting genes from the transcriptome data involved in these hormone biosyntheses could be more relevant and great addition to MS. 

Author Response

Response to comment 1).

     Figure1; We appreciate the comment and have included the description in result section in the revised version (line number 127-139). We discussed also the significance of carbohydrates in seed oil content in the discussion section (line 450 to 456)

We have taken the suggestion and added the information about the data collection in the figure legend. All the measurements of oil content, carbohydrates and hormone content were conducted from samples taken from one location in Wuhan during the season of 2016-17.

Response to comment 2).

     Table 1; The analysis for Table 1 was conducted with the data of seed tissue at 23 DAP stage as indicated in the title. At this stage, the difference in oil content became more significant between HOCL and LOCL (Figure 1).  The legends were modified to make the presentation clearer in the whole manuscript. We did consider analysis as suggested with the MapMan tool. However, the software can only run two sets of data at one time and thus it is not feasible to run such analysis unless the data at the different stage were pooled. To avoid such a potential problem, we did separately MapMan analysis with each stage (supplementary figure 1 and 2).

Response to comment 3).

    Figure 6; As most of the transcriptomic profiling analyses, qRT-PCR assay serves as a verification of RNA-Seq data. We took genes and samples randomly for such a validation because this way has some advantages in eliminating possible systematic errors. Such a sampling procedure is also popular in references to similar types of study. It is also understandable that it will be not practical to run qRT-PCR to verify genes in all different pathways with such a large volume of transcriptomic data. 

Response to comment 4).

     Figure 11; The suggestion is taken. More details about the protocol for hormone quantification have been supplied in the section of materials and methods. We had the analysis in Figure 10 and supplementary table 2.

Round  2

Reviewer 1 Report

The manuscript has been improved and now it is suitable for  the pubblication.